# Learning Language-Conditioned Robot Behavior from Offline Data and Crowd-Sourced Annotation

**Suraj Nair**[1], **Eric Mitchell**[1], **Kevin Chen**[1], **Brian Ichter**[2], **Silvio Savarese**[1], **Chelsea Finn**[1,2]
[1]Stanford University, [2]Robotics at Google

**Abstract:** We study the problem of learning a range of vision-based manipulation tasks from a large offline dataset of robot interaction. In order to accomplish this, humans need easy and effective ways of specifying tasks to the robot. Goal images are one popular form of task specification, as they are already grounded in the robot's observation space. However, goal images also have a number of drawbacks: they are inconvenient for humans to provide, they can over-specify the desired behavior leading to a sparse reward signal, or under-specify task information in the case of non-goal reaching tasks. Natural language provides a convenient and flexible alternative for task specification, but comes with the challenge of grounding language in the robot's observation space. To scalably learn this grounding we propose to leverage *offline robot datasets* (including highly sub-optimal, autonomously collected data) with *crowd-sourced natural language labels*. With this data, we learn a simple classifier which predicts if a change in state completes a language instruction. This provides a language-conditioned reward function that can then be used for offline multi-task RL. In our experiments, we find that on language-conditioned manipulation tasks our approach outperforms both goal-image specifications and language conditioned imitation techniques by more than 25%, and is able to perform visuomotor tasks from natural language, such as "open the right drawer" and "move the stapler", on a Franka Emika Panda robot.

**Keywords:** Natural Language, Offline RL, Visuomotor Manipulation

## 1 Introduction

We are motivated by the goal of generalist robots which can be commanded to complete a diverse range of manipulation tasks. Doing so requires humans to be able to effectively specify tasks for the robot to solve. One popular approach to task specification is through goal-states, which by definition are grounded in the robot's observation space, making them a natural choice for self-supervised techniques [1, 2, 3, 4]. However, goal-state specification comes with a number of drawbacks, including (a) human effort required in generating a goal state to provide the robot, (b) task over-specification resulting in a sparse reward signal (e.g. a goal image for the task of pushing a *single* object also specifies positions of all other objects and the robot itself), and (c) task under-specification (e.g. moving to the right indefinitely). Natural language presents a promising alternative form of specification, providing an easy way for humans to communicate tasks. Moreover, natural language can flexibly represent non-goal reaching tasks and tasks with varying degrees of specificity, such

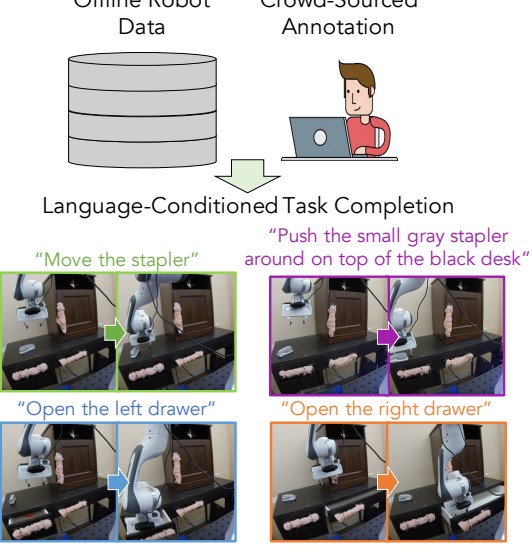

Figure 1: We learn language-conditioned visuomotor policies using sub-optimal offline data, crowd-sourced annotation, and pre-trained language models, enabling a real robot to complete language-specified tasks while being robust to complex rephrasings of the task description.

5th Conference on Robot Learning (CoRL 2021), London, UK.

as grasping *any* marker from a desk with several markers, while a single goal image could only capture one instance of success. To this end, we study the problem of learning *language-conditioned visuomotor manipulation skills from offline datasets of robotic interaction.*

Despite the abundant benefits of being able to command robots with natural language, such agents have remained out of reach. A major challenge in acquiring such agents is that the language instructions need to be grounded in the agent's high-dimensional observation space. Learning this grounding is difficult, and requires diverse interaction data paired with language annotations. Recent works have made progress towards learning such grounding by annotating data collected by humans [5, 6, 7]; however, collecting many human teleoperated trajectories on real robots can be costly and time consuming, and is thus difficult to scale to a broad set of language conditioned behaviors.

Our key insight is that a practical and scalable way to ground language is to combine autonomously-collected offline datasets of robotic interaction with post-hoc crowd-sourced natural language labels. Unlike prior work, we do not assume this data comes from a human expert or contains optimal actions, allowing the agent to leverage a wide range of data sources such as autonomous exploration data (e.g. random, scripted, intrinsically motivated), replay buffers of trained RL agents, human expert data (e.g. demonstrations, human play), and data without action labels. Given such pre-collected data, we can then use crowd-sourcing to scalably label trajectories with natural language labels describing the behaviors in the data. To learn from this sub-optimal data with noisy annotations, we learn a classifier which takes as input a natural language instruction and an initial and final image, and predicts whether or not the transition completes the instruction. This learned classifier can then be used as a language-conditioned reward for offline RL to learn language-conditioned behaviors.

Concretely, in this work we propose to learn language conditioned skills from vision using sub-optimal, autonomously-collected offline data and crowd-sourced annotations (See Figure 1). We present a simple technique to learn language-conditioned rewards from this data, which we call **L**anguage-conditioned **O**ffline **Re**ward **L**earning (LOReL), and combine it with visual model-predictive control to complete language conditioned tasks (See Figure 2). In our experiments in simulation, we observe that even with data collected by a random policy our proposed method solves language-conditioned object manipulation tasks 25% more effectively than language conditioned imitation learning techniques, as well as $\sim 30\%$ more effectively than goal-image conditioned comparisons which over-specify the task. Additionally, we observe that by virtue of leveraging pretrained language models our learned reward is capable of generalizing from scripted language instructions to unseen natural language zero-shot, suggesting that knowledge in pretrained language models can enable more efficient learning of grounded language as observed in prior work [6, 8]. Finally, we leverage an existing real robot dataset of sub-optimal data, label the dataset using crowd-sourcing, and use it to complete five visuomotor tasks specified by natural language, such as "open the right drawer" or "move the stapler" on a real Franka Emika Panda robot.

## 2   Related Work

There is a rich literature of work which studies interactive agents, and grounding their behaviors in language [9, 10, 11, 12]. Many prior works have studied this problem in the context of *instruction following*, where an agent aims to complete a task specified by formal language/programs [13, 14, 15, 16, 17, 18] or natural language [10, 11, 19, 20, 21, 22]. While these approaches have been largely studied in simulated spatial games [19, 23, 24, 25] or in object-directed visual navigation in simulated robots [26, 27, 28, 29, 30, 31, 25] some of which include high-level object interaction [32], in this work we focus on the domain of learning control for *vision-based robotic manipulation*.

Early works have approached instruction following with strategies like semantic parsing mapped to motion primitives or pre-defined actions to execute tasks in virtual domains [33, 34, 35, 36] and on mobile robots [37, 38]. Like our approach these methods don't require expert demonstrations; however unlike these approaches, we directly learn robotic control from images and natural language instructions, and don't assume any predefined motion primitives. More recently, end-to-end deep learning has been used to condition agents on natural language instructions [39, 26, 40, 29, 41, 6, 7, 42], which are then trained under an imitation and/or reinforcement learning objective. In the reinforcement learning setting, works have adopted a range of strategies, from language-conditioned reinforcement learning while leveraging environment rewards [23, 43, 44, 45, 46, 47] to using language as a reward bonus to densify the environment reward and aid in exploration [48, 49, 50, 51, 52]. In contrast, we do not assume any environment provided reward signal, and rather aim to *learn* effective language-conditioned rewards from annotated data of interaction.

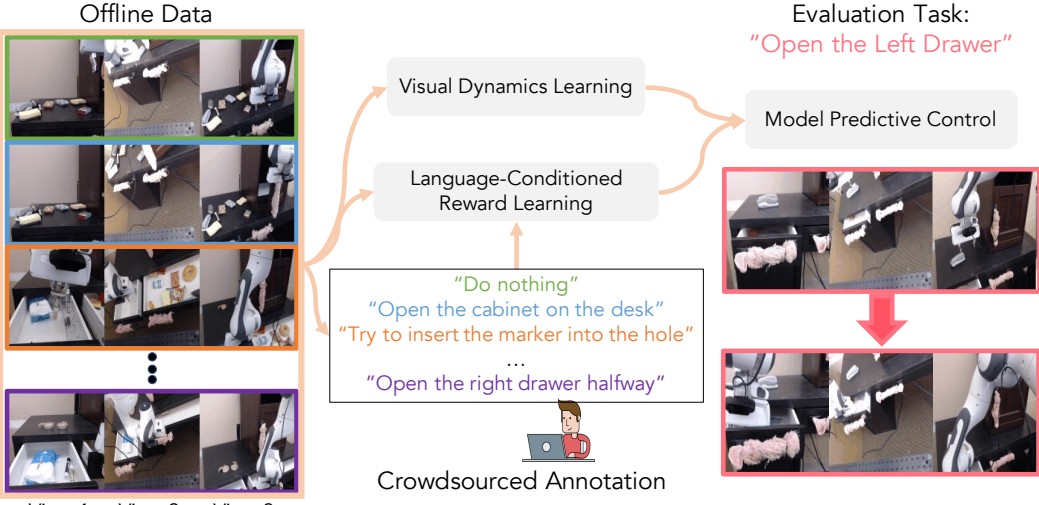

Figure 2: **Language-conditioned Offline Reward Learning (LOReL)**. We propose a technique to learn language-conditioned behavior from offline datasets of robot interaction (**left**). To do so, we crowd-source natural language annotations describing the behavior in the offline data, and use it to learn a language-conditioned reward function (**middle**). We then combine this reward and a learned visual dynamics model through model predictive control to complete language specified tasks form vision on a real robot (**right**).

Numerous prior works have also studied learning language conditioned rewards online from demonstrations or examples of successful completion of tasks and language annotations [5, 24, 41, 51]. These works then use the learned reward to optimize policies through online RL, often using the agents own online experience to train the reward [24, 53]. While these works also learn language-conditioned rewards, and in some cases also use discriminative techniques to learn the reward [24, 51], running language-conditioned online RL on a physical robot can be prohibitively time consuming. Our work aims to learn language-conditioned behaviors from entirely offline datasets (which may be highly sub-optimal), making it feasible to learn language-conditioned behaviors on real robots.

Other works have studied using offline data in the form of demonstrations [7] or human teleoperated trajectories (i.e. "play data") [6], to learn language-conditioned robotic agents in simulation. Most related is Lynch and Sermanet [6] who also use crowd-sourcing to annotate play data with natural language instructions. Critically, these works treat the offline data as near optimal, to the extent that behavior cloning techniques can be used to learn language-conditioned policies. Unlike these works, we don't make any assumptions about the optimality of the actions in the collected data, allowing the agent to learn from broader offline datasets, including autonomously collected data, which can be considerably easier to collect at scale on a real robot than human tele-operation data [54, 55]. Moreover, we observe in Section 5.1 that our proposed approach outperforms imitation learning techniques on such data, and in Section 5.3 that our method is effective on a real robot.

Many prior works have studied how robots can learn to complete a wide range of tasks from vision. While many approaches have been taken to task-specification, including task IDs [56, 57], robot and human demonstrations [58, 59, 60], and meta-learning from rewards [61], a common approach is goal-conditioned learning [62, 63, 2, 1], where an agent learns to reach particular goal states or distributions [64]. Many approaches have been applied to this domain, ranging from goal-conditioned model-free learning [2, 65, 57, 66] with goal relabeling [63], model-based planning with a learned visual dynamics model [67, 68], to methods which combine the both [69]. Unlike these works, the focus of this work is multi-task visuomotor learning from natural language specifications. Furthermore, we find in Section 5.1 that using our language-conditioned reward we can more effectively complete tasks than leveraging a goal image specification, while requiring less human effort to specify the task.

## 3 Preliminaries

In this work we consider an interactive agent which aims to complete $K$ tasks $\{\mathcal{T}_k\}_1^K \subset \mathcal{T}$ where $\mathcal{T}$ denotes the space of all tasks. For each task $\mathcal{T}_i \in \mathcal{T}$ the agent operates in a Markov decision process MDP $\mathcal{M} = (\mathcal{S}, \mathcal{A}, p, \mathcal{R}_i, T)$ where $\mathcal{S}$ is the state space (in our case RGB images), $\mathcal{A}$ is the robot's action space, $p(s_{t+1}|s_t, a_t)$ is the robot environment's stochastic dynamics, $\mathcal{R}_i : \mathcal{S} \times \mathcal{S} \to \{0, 1\}$

indicates the binary reward at state $s$ for completing task $\mathcal{T}_i$ from initial state $s_0$, and $T$ is the episode horizon. Lastly, let $\mathcal{L}$ denote the set of all natural language, and let $\mathcal{L}_i \subset \mathcal{L}$ denote the set of language instructions which describe task $i$. Note that there can exist many instructions $l \in \mathcal{L}_i$ which describe a task $\mathcal{T}_i$ (e.g. "pick up the blue marker" and "grasp and lift the blue marker"), and any particular instruction $l \in \mathcal{L}$ can describe multiple tasks (e.g. "pick up the marker" can describe the task of picking up the blue marker or the green marker).

In this work we assume that the true reward function $\mathcal{R}_i$ for each task $\mathcal{T}_i$ is unobserved, and must be inferred from natural language. Concretely, we assume access to an offline dataset $\mathcal{D}$ of $N$ trajectories, where each trajectory $\tau_n$ is a tuple containing a sequence of states and actions, and a single natural language instruction $\tau_n = ([(s_0, a_0), (s_1, a_1), ..., (s_T)], l_n)$. We assume that $l_n \in \mathcal{L}_i$ for at least one task $\mathcal{T}_i \in \mathcal{T}$ for which $\mathcal{R}_i(s_0, s_T) = 1$. Note $\mathcal{T}_i$ does not need to be in $\{\mathcal{T}_k\}_1^K$, meaning the offline data/annotations can consist of tasks unrelated to the robot's target tasks (e.g. "doing nothing"). Our goal then is to learn a parametrized reward model $\mathcal{R}_\theta : \mathcal{S} \times \mathcal{S} \times \mathcal{L} \to [0, 1]$ which conditioned on a language instruction $l$, initial state $s_0$, and state $s$ infers the true reward function $\mathcal{R}_i(s_0, s)$ for *some* task $\mathcal{T}_i$ for which $l \in \mathcal{L}_i$. Given $\mathcal{R}_\theta$, we aim to instantiate a stochastic language-conditioned policy $\pi : \mathcal{S} \times \mathcal{S} \times L \to \mathcal{A}$, which conditioned on a natural language instruction $l$ produces actions to maximize the expected sum of rewards $\sum_{t=0}^{T} \mathcal{R}_i(s_0, s_t)$ for the inferred task $\mathcal{T}_i$. Note that this formulation captures tasks that are reflected in a change of state, but not tasks which are path dependent (e.g. "close the drawer *slowly*").

## 4 Language-conditioned Offline Reward Learning (LOReL)

Now we describe how we go about learning our parametrized language-conditioned reward $\mathcal{R}_\theta$ from $\mathcal{D}$, as well as how we instantiate our language-conditioned policy $\pi$ to maximize the learned reward, also shown in Figure 2. Our key idea is that while we cannot make assumptions about the optimality of the behavior in $\mathcal{D}$, we can use the the initial and final states of each trajectory and provided language annotations to ground what changes in state correspond to successful completion of language instructions. Then to learn control we can leverage all of the data in $\mathcal{D}$ to learn a global task-agnostic model of the dynamics of the robots environment, which can be combined with the learned reward via model-predictive control (MPC) to complete language conditioned tasks.

### 4.1 Learning the Reward Function

Given the provided dataset $\mathcal{D} = [\tau_1, ..., \tau_N]$ of $N$ trajectories $\tau_n = ([(s_0, a_0), (s_1, a_1), ..., (s_T)], l_n)$, how might we go about learning our reward function $\mathcal{R}_\theta$? Critically, the behavior policy which collected this data could be sub-optimal. Therefore, we cannot assume the optimality of any particular action taken. However, because the human provided annotations describe the task being completed in the video, the assumption we can make about the data is that going from the start to the end of the trajectory constitutes completion of $l_n$. Therefore, we implement our reward function as a binary classifier $\mathcal{R}_\theta(s_0, s, l)$ which looks at the initial state $s_0$, current state $s$, and language instruction $l$ and predicts if going from the initial state to the current state satisfies the language instruction.

Training the reward function in this manner has numerous favorable properties. First, unlike explicitly predicting a single instance of a successful goal state for a language instruction or vice-versa, a classifier can easily capture the many-to-many mapping that exists between language instructions and tasks. Doing so allows it to capture the full space of successful behavior even in cases where there exists many possible language instructions $l$ which can describe completing a task $\mathcal{T}_i$ and many possible pairs of initial and final states $(s_0, s_T)$ which can constitute successfully completing any given instruction $l$. Second, by virtue of being *context-dependent* on the initial state, the reward function can be used in closed loop planning to perform behaviors indefinitely without additional specification (i.e. the reward for "move right" is relative to the agent's current position, so applying it iteratively will encourage the agent to continuously move right). Lastly, unlike other works which use classifiers for single-task reward learning [70, 71, 55] on robots, a language-conditioned reward classifier can flexibly represent many tasks with an easy to provide form of task-specification. Concretely, we sample positive examples $(s_0, s_T, l_t) \in \tau_n \sim \mathcal{D}$ from the annotated dataset which constitute successfully completing an instruction and generate negative examples which don't complete instructions $(s_0', s_T', l') \sim \mathcal{N}$ also from $\mathcal{D}$ (described in detail in the next sections). We then minimize the binary cross entropy loss:

$$\mathcal{J}(\theta) = \mathbb{E}_{(s_0, s_T, l) \sim \mathcal{D}}[\log(\mathcal{R}_\theta(s_0, s_T, l))] + \mathbb{E}_{(s_0', s_T', l') \sim \mathcal{N}}[\log(1 - \mathcal{R}_\theta(s_0', s_T', l'))]. \quad (1)$$

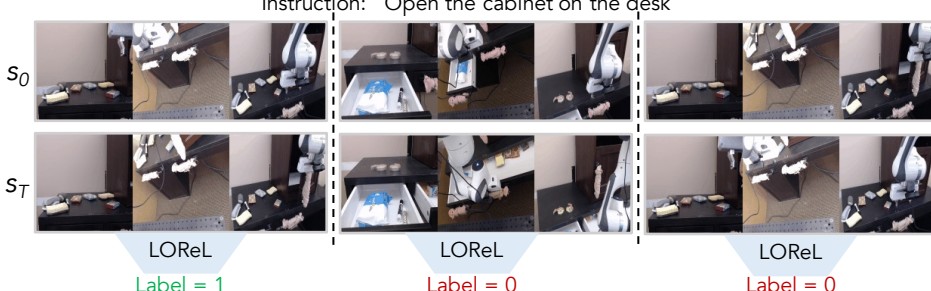

Figure 3: **Training LOReL.** We train LOReL on balanced batches of positive examples where the initial/final image transition satisfies the language command (**left**), negative examples where the initial/final states satisfy a *different* instruction (**middle**), and negative examples where the initial and final image are reversed (**right**).

**Positive Selection.** Selecting positive examples for the classifier is straightforward, as we know the initial and final state in an episode labeled with instruction $l$ satisfy that command. However, it is also highly likely that there are other states near the beginning and end of the episode for which the instruction is satisfied. Therefore we employ a noisy labeling scheme where we label any $(s_i, s_j, l)$ for which $i \leq \alpha T$ and $j \geq (1-\alpha)T$ as a positive. Higher values of $\alpha$ may occasionally include false positives, but also significantly increase the set of positives which can be used for training.

**Negative Selection.** First, we choose initial and final states from episodes with *different* language instructions as negatives. Specifically, for language command $l$, we select negatives $s'_0, s'_T$ by selecting any $(s_0, s_T, l') \sim \mathcal{D}$ where $l' \neq l$. Note that since there may be instructions $l' \neq l$ which describe the same task, this may occasionally yield false negatives, however like prior work [70, 55] we find that we can learn an effective reward despite noisy negatives. Second, to encourage the reward to capture temporal progress (as opposed to focusing on spurious visual features), we also include the example $(s_T, s_0, l)$ as a negative in training $\mathcal{R}_\theta$. Ultimately, we train on balanced batches of positive examples and both types of negative examples (See Figure 3).

**Data Augmentation.** Reward functions trained using classifiers have been shown to be prone to over-fitting creating a sparse or incorrect reward signal [55]. This issue is further exacerbated by the fact that we only have limited positive examples per episode. To combat this, we use visual data augmentation in the form of affine transformations and color jitter as well as uniform noise in the embedding space of language instructions to prevent classifier over-fitting.

**Leveraging Pre-Trained Language Models.** Finally, learning the meaning of raw natural language while simultaneously grounding the robot's actions using only crowd-sourced data of a few thousand robot episodes with language annotations poses a significant challenge. Therefore we leverage a fixed pre-trained distilBERT sentence encoder [72], to encode the natural language commands into a fixed length vector in $\mathbb{R}^{768}$ before they go into the classifier. We find in Section 5.2 that by using the pre-trained model we can generalize to unseen natural language commands from synthetic data.

### 4.2 Learning Language Conditioned Policies with Visual Model Predictive Control

Once trained, the learned reward function $\mathcal{R}_\theta$ in principle could be used with any form of offline reinforcement learning to learn language-conditioned policies. In this work, we aim to learn visuo-motor control on real robots from large datasets of sub-optimal or even random offline data. Model-based RL techniques have been particularly effective in this endeavor [67, 60], and in our case all offline data can be used to train a single task-agnostic visual dynamics model. We then use this model with planning to maximize the learned language-conditioned reward $\mathcal{R}_\theta$. Specifically, we learn a forward visual dynamics model $s_{t+1} \sim p_\theta(s_t, a_t)$ which is trained on the entire offline dataset $\mathcal{D}$, and does not use language annotations. We leverage off-the-shelf action-conditioned video prediction frameworks for learning this model [73, 74], which we describe in detail in the supplement.

Given the learned dynamics model $p_\theta$ and the learned reward function $\mathcal{R}_\theta$, we then use model predictive control to instantiate a policy to complete language-conditioned tasks. Specifically, given a language instruction $l$ and initial state $s_0$, we sample $M$ different actions sequences of length $H$, which we feed through $p_\theta$ to get a predicted future state $\hat{s}^m_{t+H}$. For each prediction we compute the reward as $\mathcal{R}_\theta(s_0, \hat{s}^m_{t+H}, l)$. Action sequences are optimized to maximize reward using the cross-entropy method (CEM) [75], until the best action sequence is applied in the environment (Figure 4).

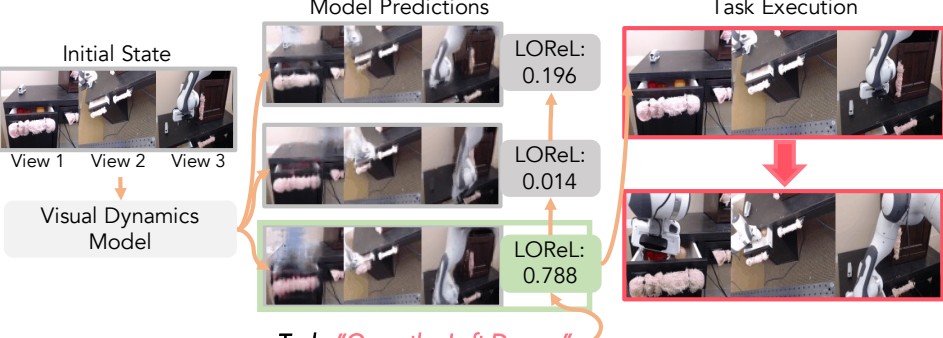

Figure 4: **Executing Language-Conditioned Policies with LOReL.** To execute language-conditioned behavior, we perform model predictive control with a learned visual dynamics model and LOReL. Specifically, from the initial state we predict many future states for different action sequences (**left/middle**). We then rank those sequences according to the LOReL reward for the user specified natural language instruction (**middle**). After multiple iterations, the best action sequence is stepped in the environment executing the task (**right**).

## 5 Experiments

In our experiments we aim to study three main questions. **(1)** How does our proposed method for learning language conditioned policies from offline data compare to both language-conditioned and goal-image conditioned prior methods? **(2)** By virtue of using pre-trained language models, to what extent can our method generalize to unseen natural language commands? **(3)** Can our method be used to solve visuomotor tasks on a real robot using crowd-sourced annotations? We study experiments (1) and (2) in simulation, and experiment (3) on a Franka Emika Panda robot positioned in front of a desk. For qualitative results and videos, please see https://sites.google.com/view/robotlorel.

### Simulated Domain

We study our first two experimental questions in a simulated domain developed on top of the Meta-World [56] environment, where a simulated sawyer robot interacts on a tabletop with a drawer, a faucet, and two mugs (see Figure 5 (left)). In this domain, we collect an offline dataset of 50,000 episodes by running a random policy in the environment, and label episodes procedurally using the true environment state yielding 2311 unique instructions (see Figure 5 (right)). After training on this data, we evaluate on 6 seen tasks which involve (1) closing the drawer, (2) opening the drawer, (3) turning the faucet left and (4) right, and (5) pushing the black mug right, and (6) pushing the white mug down.

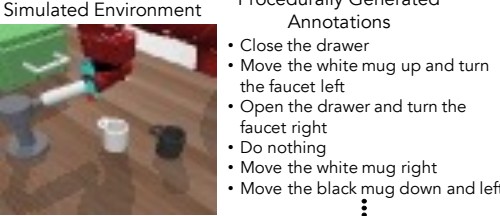

Figure 5: **Simulated Domain/Data.** We leverage a simulated domain developed on top of Meta-World [56] which contains a Sawyer robot interacting with a drawer, faucet, and two mugs (**left**). We collect data using a random policy, and annotate episodes with language instructions using the environment state (**right**).

### 5.1 Does LOReL enable effective language-conditioned behavior compared to prior work?

In this experiment we aim to evaluate how LOReL compares to prior techniques for learning language and goal image conditioned behavior on the 6 target tasks described previously.

**Comparisons.** We compare LOReL (**Ours**) to language-conditioned behavior-cloning (**LCBC**), which imitates the behavior in the offline dataset conditioned on the language instruction label, which is reflective of prior works that use imitation learning to learn language-conditioned behavior [6, 7]. We also compare to language-conditioned RL (**LCRL**), which labels the final state in each episode as having reward 1 for the annotated language instruction and 0 elsewhere, and trains a language conditioned-policy using offline Q learning, which reflects a fully offline version of the low-level policy used in [43]. Furthermore, we compare to using a goal-image as the task specification instead of language, and provide the agent with a ground truth goal-image of the object in its desired position, with which we use either L2 pixel distance (**Pixel**) or LPIPS [76] similarity (**LPIPS**) as a planning cost, reflective of prior work in visual MPC [67, 77]. Finally, we include an (**Oracle**) which uses the ground truth dynamics model and ground truth reward indicating the upper bound on the performance of the CEM planner, as well as the performance of a (**Random**) policy. All comparisons use the same architecture and data where possible; see the supplement for further details.

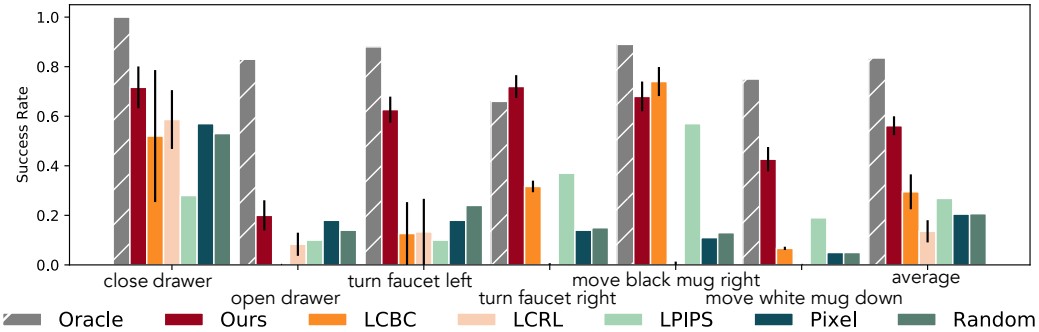

Figure 6: **Comparison to Prior Work.** On 6 simulated language-conditioned tasks, we find that LOReL (Ours) outperforms language-conditioned imitation learning (LCBC) and Q-learning (LCRL) as well as goal-image task specification (LPIPS/Pixel) by over 25%. Success rates/standard error computed over 3 seeds of 100 trials.

**Results.** Figure 6 shows the success rates over 3 seeds of 100 trials, ordered by legend. We observe first that our proposed approach outperforms the next best method, language-conditioned behavior cloning, by more than 25%. By learning a language-conditioned reward and planning over it, the robot executes the tasks more effectively than what it observed in the data. On the other hand, because the data is sub-optimal, language-conditioned imitation learning is only able to learn coarse directions associated with each task, and as a result fails on tasks that require more fine-grained motion like "turn faucet left". Second, we observe that language-conditioned RL with a binary reward struggles to learn at all, indicating the difficulty in jointly learning language-grounding and control. Finally, we find that using goal-images with a pixel cost also fails, performing comparably with a random policy. We observe that the agent tries to match the arm position in the goal instead of the interacting with the objects, highlighting the limitation of goal-images in their tendency to *over-specify* the task. Using LPIPS similarity improves performance, however is still $\sim 30\%$ worse than our method.

### 5.2 Can LOReL generalize zero-shot to unseen natural language commands?

In our second experiment, we study our methods ability to generalize to unseen instructions by nature of using pre-trained language models. Specifically, for the six target tasks, we test our method with a *rephrased instruction*, which was completely unseen during training. We evaluate task performance on the **Original** commands, on the commands with an **Unseen Verb** (e.g. "turn faucet left" → "rotate faucet left"), on the command with an **Unseen Noun** (e.g. "move black mug right" → "move dark cup right"), and **Unseen Verb+Noun** (e.g. "close drawer" → "shut cabinet"). Finally, we also test on **Unseen Natural Language** commands collected from 9 human volunteers who were asked to rephrase the command in a creative way, for example "turn faucet left" → "Spin nozzle left". The full set of unseen instructions for each task is in the supplement.

| Success Rate | LOReL | LOReL (-PM) |
|---|---|---|
| Original | **56 ± 1%** | 40 ± 1% |
| Unseen Verb | **51 ± 3%** | 33 ± 2% |
| Unseen Noun | **51 ± 1%** | 39 ± 4% |
| Unseen Verb + Noun | **47 ± 2%** | 17 ± 3% |
| Unseen Natural Language | **46 ± 2%** | 21 ± 1% |

Table 1: **Generalization to Unseen Commands.** We compare the models ability to generalize to unseen instructions and natural language when learning with a fixed, pre-trained language model (LOReL), compared to training the same architecture from scratch only on the grounded language data (LOReL (-PM)). We see significant performance improvement in both seen/unseen commands by using a pre-trained model.

In Table 1, we see that on average when changing the verb *or* noun, we only see a drop in success rate of 5%, and when changing both or using human provided natural language, we see at most a 10% drop in success rate. Furthermore, we compare performance with and without using the pre-trained language model, and observe that without the pre-trained language model performance is worse on seen instructions and drops significantly more on unseen instructions (up to 23% vs 10%), suggesting that the pre-trained language model is essential to learning and generalization, consistent with results in prior work on language-conditioned imitation [6]. This result also suggests that the ungrounded knowledge in large language models may enable learning language groundings from small datasets or *entirely programmatic language* which can generalize to natural language.

### 5.3 Can LOReL be used to learn language-conditioned visuomotor skills on a real robot?

Finally, we study the efficacy of our method in learning language-conditioned behavior on a real robot using sub-optimal offline data and crowd-sourced annotation. We consider a Franka Emika

Panda robot mounted over an IKEA desk with two drawers and a cabinet, which can hold a range of objects (Figure 7). The robot's observation space consists of 4 camera viewpoints, each providing $64 \times 64$ RGB images. The robot's action space is delta end-effector control.

**Data and Annotation.** We use an offline 3000 episode (150000 frame) dataset without any modification from concurrent work [78], which trains policies for different behaviors on the IKEA desk using online RL. As a result, our dataset consists of diverse behaviors, but is also sub-optimal in that it comes from the replay buffer of a learning policy which will often not complete any task or will complete tasks in highly sub-optimal ways.

To annotate the data, we leverage crowd-sourcing, specifically Amazon Mechanical Turk. We ask human annotators to describe the behavior, if any, that the robot is doing, and to phrase it as a command without any pre-specified template. We collect 6000 annotations, two per episode, containing a total of 1699 unique instructions, examples of which can be seen in Figure 7. We filter out episodes for which annotators wrote the robot did nothing or indicated they could not understand what the robot was doing. See supplement for details about the environment, data, tasks, distribution of annotations, and annotator interface.

Figure 7: **Robot Domain/Data.** Our real robot domain consists of a Franka Emika Panda mounted over an IKEA desk (**left**). We crowd-source annotations describing the tasks being completed in an offline dataset from this environment (**right**).

**Results.** We find in Table 2 that, by training LOReL on this annotated dataset and using it for visual MPC with a learned dynamics model in this domain, the robot can complete 5 language-conditioned skills with a 66% success rate on average across skills. Additionally, we find that removing negative training examples with the initial/final state flipped (**LOReL (-FN)**) reduces performance by 30%, suggesting that such negatives are important for the reward to capture temporal progress and prevent over-fitting to objects.

Finally, we test LOReL's robustness to more complex vocabulary and instruction length, by replacing the commands for opening the left drawer and moving the stapler with "Open the small black and white drawer on the left fully" and "Push the small gray stapler around on top of the black desk" respectively. We find that LOReL is still able to succeed **7/10** and **5/10** times respectively, providing evidence that it is robust to instruction phrasing, consistent with the results in Section 5.2.

| Task (10 Trials Each) | LOReL | LOReL (-FN) |
|---|---|---|
| "Open the left drawer" | **90**% | 30% |
| "Open the right drawer" | **40**% | 0% |
| "Move the stapler" | **50**% | 0% |
| "Reach the marker" | **70**% | **70**% |
| "Reach the cabinet" | **80**% | **80**% |
| Average over tasks | **66**% | 36% |

Table 2: **Real Robot Results.** Using LOReL we are able to complete 5 language-conditioned skills on a robot specified by language with a 66% success rate.

## 6   Limitations and Future Work

We have presented LOReL, a technique for learning language conditioned behavior from offline data and crowd sourced annotation that is effective for visuomotor control on real robots and is capable of generalizing to unseen language instructions. However, a number of limitations remain. First, in its current form LOReL can only capture tasks which are reflected in some change of state, but cannot capture tasks which are path dependent (e.g. "move in a circle slowly"). One exciting direction for future work to address this is to train LOReL on full video clips. Second, while in this work we have focused on learning short-horizon skills from language, composing these skills to solve long-horizon language-specified tasks is important for making robots useful in the real world. Data sources with long-horizon behaviors, and more powerful planners and visual dynamics models are necessary to enabling these longer horizon tasks. Finally, while we have presented language specification as an alternative to goal images, goal images maintain the benefit of being self-supervised and in some cases can be effective task specification. Unifying both forms of specification for robots would be a valuable future direction.

**Acknowledgments**

The authors would like to thank Bohan Wu for valuable input throughout the project and assistance in conducting the robot experiments. The authors would also like to thank members of the IRIS and RAIL labs for providing valuable feedback. Suraj Nair is funded in part by an NSF GRFP. Eric Mitchell is funded by a Knight-Hennessy Fellowship. The authors would also like to thank the AMT annotators who helped label the robot data. Toyota Research Institute ("TRI") provided funds to assist the authors with their research but this article solely reflects the opinions and conclusions of its authors and not TRI or any other Toyota entity.

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
