# OpenReview forum: "Learning Language-Conditioned Robot Behavior from Offline Data and Crowd-Sourced Annotation"
_robot-learning.org/CoRL/2021/Conference — CoRL2021 Poster_

### Official Review · Reviewer_avX2 · 2021-07-03

**Originality:** Good
**Technical Quality:** Very Good
**Clarity Of Presentation:** Excellent
**Impact:** 4

**Recommendation:**

Weak Accept: I recommend accepting the paper, but will not argue for my recommendation if the majority of other reviewers have a different opinion.

**Summary:**


This paper addresses the problem of learning to follow natural language instructions to solve vision-based manipulation tasks. Specifically, it assumes the availability of a large offline dataset of robot interaction that could be sub-optimal or even random with crowd-sourced descriptions that describe robot's behaviors. To solve this problem, this paper proposes to learn a classifier from the offline dataset that takes an initial state, a final state, and a description of the trajectory as input and outputs if the final state fulfills the description as well as learn a forward visual dynamics model (an action-conditioned video prediction model). To learn an effective classifier, many techniques are proposed, including learning from negative samples, data augmentation, and employing a pre-trained language model. During testing, the proposed framework rollouts action sequences with the learned visual dynamics model and evaluate them based on the reward provided by the classifier conditioned on the task language instruction and the predicted final states. Then, the action sequence that maximizes the classifier reward is selected to be executed. The proposed framework is compared against a variety of baselines that use a different task specification (i.e. a goal image) or leverage the offline data differently (e.g. multi-task RL or imitation learning). The experiments in simulation and on a real Franka Emika Panda robot show that the proposed framework outperforms baselines and can reasonably generalize to unseen natural language instructions. I believe this work studies a promising research direction (i.e. natural language instruction following with offline datasets), proposes an interesting framework, introduces effective techniques, and presents comprehensive evaluations. Therefore, I would like to see this work presented in CoRL.

**Issues:**


Described in the strengths and weaknesses section.

**Reviewer Expertise:**

Good: General knowledge of the area

**Strengths And Weaknesses:**


## Paper strengths and contributions

**Motivation and intuition**
- This paper gives clear and organized reasons why representing tasks using goal states could be problematic.
- Leveraging natural language descriptions to instruct robots is easy and natural especially for non-expert users and flexible.
- Employing an offline dataset could potentially be more sample efficient. Also, learning a binary classifier should require much less data than multi-task RL and imitation learning.

**Technical contribution**
To effectively learn a binary classifier that can reliably predict if the transition between a pair of initial and final states align with a natural language description, this paper proposes several techniques, including:
- Leveraging intermediate states (close to initial or final states) to obtain more positive examples.
- Defining and sampling negative examples with mismatched initial-final state pairs and language instructions as well as swapped initial-final state pairs. The effectiveness is shown by comparing LORL to LORL (-Neg) in Table 2.
- Employing data augmentation in the image state space (e.g. affine transformations and color jitter) and the latent space of the language instruction (i.e. noise).
- Using a pre-trained language model (a Distilbert sentence encoder). This allows generalization to unseen instruction as shown in Table 1 and Figure 7.

**Clarity**
The writing is amazingly clear and the paper organization is easy to follow. I would even say this paper has the best writing among all the papers (from five major conferences) that I have reviewed so far this year. Reading this paper is enjoyable. I really appreciate the authors' efforts in polishing the writing.

**The amount of the offline data required**
In my opinion, learning a binary classifier should require way fewer data compared to learning a multi-task RL policy or imitation learning. Therefore, I believe the idea of learning just a classifier instead of a policy from the offline dataset can reduce the amount of required data. Is this the main reason why the proposed framework outperforms the baselines that also learn from language instructions?

**Annotator expertise for collecting offline datasets**
To create an offline dataset that contains robot trajectories and their descriptions, one intuitive way would be first generating language instructions that describe certain desired behaviors and asking human workers to control robots to fulfill the instructions. This requires not only access to robots but also the workers' expertise in controlling robots. Therefore, it is less practical to collect datasets this way on a large scale.

On the contrary, this paper instead first collects robot trajectories and then asks annotators to provide descriptions narrating the trajectories. This greatly alleviates the expertise needed for the annotators. Yet, producing meaningful behaviors can still be challenging.

**Ablation study**
The proposed framework employs several techniques and the provided ablation studies justify the effectiveness of some of them, including:
- The use of negative examples with mismatched initial-final state pairs and language instructions as well as swapped initial-final state pairs, shown in Table 2.
- The use of a pre-trained language model, shown in Table 1 and Figure 7.

**Experimental results**
- The presentation of the experimental results is clear.
- The proposed framework is evaluated in both a simulated environment and with a real Franka Emika Panda robot.
- The baselines are sufficient
    - To verify that the issues of representing tasks using goal states/images), a baseline (Pixel) that uses a ground truth goal-image is considered.
    - To compared against different ways of leveraging offline datasets, two baselines are considered:
        - Language-conditioned behavior-cloning (LCBC): leverages the offline datasets by performing imitation learning with language instructions.
        - Language-conditioned RL (LCRL): a multi-task RL policy that conditions on language instructions and learns from sparse reward.
- The proposed framework outperforms those baselines by a large margin.

**Reproducibility**
Given the clear description in the main paper and the details provided in the appendix, I believe reproducing the results is possible.


## Paper weaknesses and questions

**Initial-final state pairs or videos**
To evaluate if a behavior aligns with a language description, this paper proposes to represent the behavior using an initial-final state pair.
Yet, doesn't this also have the issue of representing a goal using an image (e.g. task over-specification and under-specification)? For example, the task of rotating a faucet indefinitely (mentioned in the introduction) can't be dealt with using initial-final state pairs. Therefore, I wonder if it would be better to represent a behavior using the entire video.

**Pixel baseline**
- Pixel uses L2 pixel distance as a planning cost. I wonder if it would work better using a learned latent distance (even it needs extra learning to learn a latent space) or perceptual loss.
- To verify if representing a goal using an image has the task over-specification, it would make sense to use a masked goal image to only show the parts of the image that are relevant to the task. If this masked goal baseline performs better, the task over-specification does exist.

**Learning from nonmeaningful behaviors**
This paper mentions many times that the proposed framework can learn from nonmeaningful behaviors (i.e. trajectories collected from random policies). I have a few questions regarding this statement as follows.
- L314-317 "We filter out episodes for which annotators wrote “do nothing” or indicated they could not understand what the robot was doing.": why is it necessary to filter those data then?
- Aren't data collected from random policies mainly contain those "do nothing" behaviors?
- So, my guess is that the classifier would not learn well if the majority of the offline dataset is nonmeaningful due to its imbalance. However, this is what you would get from a random policy. If this is the case, I think the authors oversell this a little bit and I would suggest the authors tone down and just claim that it can learn from sub-optimal data but not random data. Since most random data might not even contain interactions with objects, so it still needs to assume that behaviors are most meaningful in some ways.
- Following the previous point, I would like the know the performance of the model if those data are not filtered out.

**Performance of the classifier**
The experimental results only present the performance of the full pipeline of the proposed framework. I believe it would be informative to also evaluate the performance of the classifier alone.
- I suggest the authors split the offline dataset into training/validation/testing datasets so that the training dataset can be used to train the classifier, the validation dataset can be used to determine the hyperparameters, and the testing dataset can be used to evaluate the performance of it.
- In this case, it would be easier to conduct ablation studies to verify if those techniques employed to train the classifier are effective (see below).

**Missing ablation study**:
This paper employs several techniques to train the classifier, but some ablation studies are missing to evaluate if those techniques are useful.
- The noisy labeling scheme is used to create more positive examples.
- There are two types of negative examples introduced to train the classifier, including mismatched initial-final state pairs and language instructions as well as swapped initial-final state pairs. It would be informative to see which one contributes more to the performance gain.

**Collecting the offline dataset**
Is some sort of instruction template used for collecting instructions? This is probably the case so that 50,000 episodes only share 2311 unique instructions and this is also why there are experiments on generalizing from scripted language instructions (probably instructions templates) to unseen natural language. If this is the case, the authors should explicitly mention this in the paper.

**The amount of the offline data required for the multi-task RL policy and the behavioral cloning model**
As mentioned earlier in the strengths section, I assume that learning a binary classifier requires fewer data compared to learning a multi-task RL policy or a behavioral cloning model conditioned on language instructions. Therefore, I suspect that the performance gap might be attributed to insufficient data for learning the multi-task policy and BC model. To verify that, we can increase the data in the offline dataset and see if the performance of the proposed framework is the same yet the performance of the multi-task policy and BC model increases.

**Related work**
It would make the related work section more comprehensive by including some prior works that explore using formal languages (i.e. programs) as a task representation, including
- Modular multitask reinforcement learning with policy sketches
- Zero-Shot Task Generalization with Multi-Task Deep Reinforcement Learning
- Programmable agents
- Program Synthesis Guided Reinforcement Learning
- Program Guided Agent
- Hierarchical Program-Triggered Reinforcement Learning Agents For Automated Driving
- Reinforcement Learning of Implicit and Explicit Control Flow in Instructions


**Summary Of Recommendation:**


I believe this work studies a promising research direction (i.e. natural language instruction following with offline datasets), proposes an interesting framework, introduces effective techniques, and presents comprehensive evaluations. While I still have some minor concerns and questions (see above), I would like to see this work presented in CoRL. However, I would consider myself more a machine learning and reinforcement learning guy rather than robotics so my evaluations mainly come from learning perspectives.

---

> ### Author Response · Authors · 2021-08-24
> **Response to Reviewer avX2 (1/3)**
>
> Thank you for your detailed feedback. We respond to your main comments individually below, and have made revisions to the paper (highlighted in *green*) based on your comments, which we believe has improved the paper. Please let us know if you have any remaining concerns or questions!
>
> > Therefore, I believe the idea of learning just a classifier instead of a policy from the offline dataset can reduce the amount of required data. Is this the main reason why the proposed framework outperforms the baselines that also learn from language instructions?
>
> This is an interesting point. We suspect the simpler optimization objective is likely why LORL outperforms the language-conditioned RL method. On the other hand, LORL outperforms the language-conditioned imitation learning baseline because imitation learning assumes a certain level of optimality in the actions, which a classifier does not require; in particular, this assumption does not hold for the lower-quality data considered in the experiments.
>
> > Initial-final state pairs or videos To evaluate if a behavior aligns with a language description, this paper proposes to represent the behavior using an initial-final state pair. Yet, doesn't this also have the issue of representing a goal using an image (e.g. task over-specification and under-specification)? For example, the task of rotating a faucet indefinitely (mentioned in the introduction) can't be dealt with using initial-final state pairs. Therefore, I wonder if it would be better to represent a behavior using the entire video.
>
> We revised the paper to clarify that LORL can only capture tasks which are reflected in a change of state. Unlike single goal states, this can (a) capture tasks where many possible states (or changes in state) imply task completion (addressing overspecification), and can (b) be iteratively applied in a closed loop to perform task indefinitely without additional specification (e.g. the reward for “move right” is relative to the agent’s current position, so applying it iteratively will encourage the agent to continuously move right). LORL **cannot** be applied to tasks which are path dependent, e.g. “jump around in place”, and “move right slowly”. We expanded the discussion of limitations in the paper, including a discussion of using the full video clip in LORL as an extension in future work to handle such path dependent tasks.
>
> > Pixel uses L2 pixel distance as a planning cost. I wonder if it would work better using a learned latent distance (even it needs extra learning to learn a latent space) or perceptual loss.
>
> This is an interesting suggestion - we implemented this suggestion and found that using an LPIPS perceptual similarity score instead of L2 pixel distance did improve the performance of the goal image conditioned method from **21% to 27%**. While it is still considerably worse than LORL (**56%**), the perceptual loss does appear to better capture objects and their positions. We added this comparison to the paper.
>
> > To verify if representing a goal using an image has the task over-specification, it would make sense to use a masked goal image to only show the parts of the image that are relevant to the task. If this masked goal baseline performs better, the task over-specification does exist.
>
> This is another interesting idea - though is somewhat challenging to implement in our domain. However, the qualitative behavior of the agent when given a goal image is to ignore the target object and to match the arm position, which does suggest over-specification is the issue. For example when using pixel L2 distance, in 235/300 trials for “move black mug right” the agent did not make contact with the black mug at all. Further, very recent work has also explored masking the goal image in visual control and has found it to improve performance [Hu et al.] supporting the overspecification hypothesis.
>
> [Hu et al.] Know Thyself: Transferable Visuomotor Control Through Robot-Awareness. Arxiv 2107.09047.

---

> > ### Author Response · Authors · 2021-08-24
> > **Response to Reviewer avX2 (2/3)**
> >
> > > L314-317 "We filter out episodes for which annotators wrote “do nothing” or indicated they could not understand what the robot was doing.": why is it necessary to filter those data then?
> > Aren't data collected from random policies mainly contain those "do nothing" behaviors?
> > So, my guess is that the classifier would not learn well if the majority of the offline dataset is nonmeaningful due to its imbalance. However, this is what you would get from a random policy. If this is the case, I think the authors oversell this a little bit and I would suggest the authors tone down and just claim that it can learn from sub-optimal data but not random data. Since most random data might not even contain interactions with objects, so it still needs to assume that behaviors are most meaningful in some ways. Following the previous point, I would like the know the performance of the model if those data are not filtered out.
> >
> > A few important clarifications :
> > 1. While the simulation experiments used data collected by a random policy, the real robot experiments used the replay buffer of a previously trained RL agent from a previous paper. We agree the data needs to include non-trivial interactions, and the focus of the work is **not** on using exclusively random data, but rather not making assumptions about the optimality of the actions taken in the offline data. As a result, LORL can learn from data generated by a wide range of behavior policies, including
> >     -  autonomous exploration data (e.g. random/scripted/intrinsically motivated)
> >     - replay buffers from previously trained RL agents
> >     - data collected by human experts (e.g. demonstrations/play data),
> >     - or even data which does not have action labels at all (for training the reward function).
> >
> > We have revised the paper to make this more clear, including toning down the emphasis on random data and focusing more on the lack of assumptions about optimal actions as suggested.
> >
> > 2. In the simulation experiments, all episodes labeled “do nothing” are retained, which makes up ~20% of the dataset. LORL is still able to perform effectively as demonstrated by the experiments in Section 5.1 and 5.2.
> >
> > 3. In the real robot domain, the language annotations are crowdsourced using Amazon Mechanical Turk. In many cases annotators could not tell what the robot was doing, or wrote “do nothing” when they were unsure, making these labels particularly noisy. As a result, we filtered out these episodes primarily as a way of cleaning out noise in the data. We re-trained our model on the full dataset and as expected given the noisier dataset, observed a reduction in final performance (**66% -> 26%**). While our simulation results suggest that our method can handle some behavior that completes no task (~20%), in the extreme case where almost half the data is labeled as not doing a task (as is the case in the noisy real robot data), it can make learning an effective reward more difficult. Specifically, episodes of "do nothing" may dominate training batches and negative examples, making it harder to learn the difference between different language instructions, especially instructions that are infrequent in the data. In practice, post-hoc cleaning/filtering of the data requires little to no extra supervision, and can make training the reward considerably easier. We have added this result and discussion to the appendix.
> >
> > > I suggest the authors split the offline dataset into training/validation/testing datasets so that the training dataset can be used to train the classifier, the validation dataset can be used to determine the hyperparameters, and the testing dataset can be used to evaluate the performance of it.
> >
> > We do evaluate the classifier on train and validation data to tune hyperparameters and choose the trained models for control. Interestingly, the performance of the classifier alone often does not directly correlate with its performance when used for planning, as particular errors, like false positives, can be especially damaging to planning performance. We have included the learning curves for training the models in the appendix.
> >
> > > Missing ablation study: The noisy labeling scheme is used to create more positive examples.
> >
> > We use the noisy positives labeling scheme to generate additional positive training samples particularly when data is limited. As a result we leverage it on the real robot setup, but not in simulation. We have since run this ablation on the real robot and found removing noisy positives on the robot reduces performance (average success rate **66% -> 32%**), suggesting that with limited data, generating additional positives with the noisy labeling is important. We have added this result to the appendix.

---

> > > ### Author Response · Authors · 2021-08-24
> > > **Response to Reviewer avX2 (3/3)**
> > >
> > > > Missing ablation study: There are two types of negative examples introduced to train the classifier, including mismatched initial-final state pairs and language instructions as well as swapped initial-final state pairs. It would be informative to see which one contributes more to the performance gain.
> > >
> > > Regarding the two types of negative examples, the real robot ablation shows that the flipped negative examples are important to performance (**66% -> 36%** success rate). The primary role of the flipped negatives are to prevent overfitting to objects in the scene, and to capture temporal progress. In simulation since the scene is unchanged, we see limited impact of removing the flipped negatives (**56% -> 55%**) On the other hand, the mismatched negatives are the main way the reward learns to capture the effect of language, and removing them has a  drastic impact in simulation (**56% -> 27%**). We have added this result in the appendix.
> > >
> > > > Collecting the offline dataset Is some sort of instruction template used for collecting instructions? This is probably the case so that 50,000 episodes only share 2311 unique instructions and this is also why there are experiments on generalizing from scripted language instructions (probably instructions templates) to unseen natural language. If this is the case, the authors should explicitly mention this in the paper.
> > >
> > > In simulation the language annotations are generated procedurally by using the simulation state, and hence use a template. On the real robot, the language annotations are crowdsourced, and do not consist of any instruction templates. Please see the appendix to see the instructions that were given to the crowdsourced annotators. We have clarified this in the revision.
> > >
> > > > As mentioned earlier in the strengths section, I assume that learning a binary classifier requires fewer data compared to learning a multi-task RL policy or a behavioral cloning model conditioned on language instructions. Therefore, I suspect that the performance gap might be attributed to insufficient data for learning the multi-task policy and BC model. To verify that, we can increase the data in the offline dataset and see if the performance of the proposed framework is the same yet the performance of the multi-task policy and BC model increases.
> > >
> > > While we have not yet had the chance to run a thorough data size ablation, we suspect that the performance gaps are not due to data size, but rather optimization objectives and assumptions. For example, our method also learns a visual dynamics model using the same dataset, which does require reasonably large amounts of data (similar to offline RL algorithms). Rather, we suspect the poor performance of the offline model-free RL is due to a more challenging optimization of having to jointly learn control and the grounding of language, while our method disentangles these components. The poor performance of behavior cloning is due to its assumption that the actions in the data are near optimal, which is not true for the data in both the simulation and real robot experiments.
> > >
> > > > Related work It would make the related work section more comprehensive by including some prior works that explore using formal languages (i.e. programs) as a task representation,...
> > >
> > > Thank you for suggesting the relevant related work; we have added discussion of these works to the revision.

---

> > > > ### Comment · Reviewer_avX2 · 2021-08-24
> > > > **Re: Response to Reviewer avX2 (3/3)**
> > > >
> > > > I appreciate the authors' response and the effort put into improving the paper. My concerns are addressed and questions are answered.

---

> > > ### Comment · Reviewer_avX2 · 2021-08-24
> > > **Re: Response to Reviewer avX2 (2/3)**
> > >
> > > Toning down the claim makes sense. The ablation study is nice.

---

> > ### Comment · Reviewer_avX2 · 2021-08-24
> > **Re: Response to Reviewer avX2 (1/3)**
> >
> > Thanks for the explanation and for doing the additional experiment! I believe incorporating the LPIPS perceptual similarity result and discussion on learning from masked goal images would improve the paper.

---

### Official Review · Reviewer_aTgV · 2021-07-21

**Originality:** Good
**Technical Quality:** Very Good
**Clarity Of Presentation:** Very Good
**Impact:** 3

**Recommendation:**

Weak Accept: I recommend accepting the paper, but will not argue for my recommendation if the majority of other reviewers have a different opinion.

**Summary:**

This paper presents an offline multi-task reinforcement learning (RL) framework for language-conditioned tasks. First, exploration data is collected with a robot through random and often suboptimal interactions. Then this data is annotated with crowdsourced natural language labels that describe what was achieved, if anything at all. These labels are used to train a reward classifier which is subsequently used for model predictive control to achieve language-specified instructions during test time. Experimental results in simulation show that the  language-conditioned policies outperform image-goal baselines by 25%, and can even handle some unseen paraphrases of instructions. The method is also evaluated on a real-robot setup.


**Issues:**

Minor
- What was the action-space used for data-collection and execution? SE-3? SE-2? Were there any hand-crafted heuristics for the exploration process?
- How long did the random-exploration data collection process take per task?


**Reviewer Expertise:**

Very good: Comprehensive knowledge of the area

**Strengths And Weaknesses:**

Strengths:
- Overall, using language to condition policies instead of high-dimensional image-goals is an apt formulation. From an user-interaction perspective, asking users to specify image-goals is restrictive. Often the user might be expected to solve the manipulation task before the robot is asked to do so in order to generate a target image for the model, which is unreasonable. Language is also effective at capturing the semantic essence of the task on what needs to be achieved without having to extract this high-level information from images.
- Prior works in language-conditioned RL often assume access to extensive amounts of language annotations to guide the policies during rollouts. The sample inefficiency of online RL methods are at odds with collecting annotations, which is expensive and time-consuming. In this work, the language labels are instead collected in an offline fashion to circumvent this issue.
- Generally, the paper is well written and the method is easy to understand. The simulated experiments include a good set of baselines and comparisons. The method is also demonstrated on a real-robot.


Weaknesses:
- While the offline setting is beneficial for collecting language-labels, it’s also restrictive in certain ways. Initially, the robot starts off with random and highly-suboptimal behavior to collect data. With random exploration, it’s unlikely that the robot will generate meaningful trajectories that can be used to fully-exploit the compositional nature of language instructions and the ability to sequentially chain commands for long-horizon tasks. Consider the task of “setting the table” where the robot has to gather several objects (like plates and forks) and arrange them in a specific manner. This is unlikely to emerge through random exploration, and thus the language-goals in this framework are mostly restricted to quick actions like “open the drawer”.
- Another concern is also the scalability of the approach for real-world applications. The real-robot experiments in Section 5.3 used a dataset of 3000 episodes (150000 frames) for a single desk environment. While the approach was tested with unseen paraphrases, in all the evaluations the training and testing environments, i.e. the IKEA desk, were identical, as is in a typically RL setting. It’s unclear if the reward-function is just learning a one-to-one mapping from language to prior states seen during training, or it’s actually learning something broadly applicable to other settings beyond the single desk. If the reward function is indeed restricted to the specific desk environment, then scaling this approach to a wide-range of real-robot tasks would be very difficult.



**Summary Of Recommendation:**

This paper presents an interesting multi-task offline-RL method for language-conditioned tasks. While the method circumvents the issue of collecting expensive online language-annotations, it’s restricted to a specific environment and skills learned through random exploration.

**Post Rebuttal**
I thank the authors for the follow-up responses. It's great that the authors plan on releasing the dataset. Unlike WikiText, I am not sure how reusable this dataset would be without the physical setup (robot, objects, and scene). However, the simulated setup should certainly help in reproducing the results. Overall, I am still positive about the work, and will keep my score unchanged.

---

> ### Author Response · Authors · 2021-08-24
> **Response to Reviewer aTgV**
>
> Thank you for your valuable feedback.  We respond to your main comments individually below, and have made revisions to the paper (highlighted in *green*) based on your comments, which we believe has improved the paper. Please let us know if you have any remaining concerns or questions!
>
> > While the offline setting is beneficial for collecting language-labels, it’s also restrictive in certain ways. Initially, the robot starts off with random and highly-suboptimal behavior to collect data. With random exploration, it’s unlikely that the robot will generate meaningful trajectories that can be used to fully-exploit the compositional nature of language instructions and the ability to sequentially chain commands for long-horizon tasks. Consider the task of “setting the table” where the robot has to gather several objects (like plates and forks) and arrange them in a specific manner. This is unlikely to emerge through random exploration, and thus the language-goals in this framework are mostly restricted to quick actions like “open the drawer”.
>
> One important clarification - while the paper used a random exploration policy to collect data in simulation, the real robot experiments used the replay buffer of a previously trained RL agent. Critically, the distinction of our method to prior work is **not** in using specifically random exploration data, but rather a method which does not make assumptions about the optimality of the actions taken in this offline data. As a result, LORL can learn from data generated by a wide range of behavior policies, including
> 1. autonomous exploration data (e.g. random/scripted/intrinsically motivated)
> 2. replay buffers from previously trained RL agents
> 3. data collected by human experts (e.g. demonstrations/play data),
> 4. or even data which does not have action labels at all (for training the reward function).
>
> If the offline data does consist of long-horizon behaviors, then LORL should be able to learn to capture them. We have revised the paper to more clearly explain the data sources LORL can leverage in the revision.
>
> > Another concern is also the scalability of the approach for real-world applications. The real-robot experiments in Section 5.3 used a dataset of 3000 episodes (150000 frames) for a single desk environment. While the approach was tested with unseen paraphrases, in all the evaluations the training and testing environments, i.e. the IKEA desk, were identical, as is in a typically RL setting. It’s unclear if the reward-function is just learning a one-to-one mapping from language to prior states seen during training, or it’s actually learning something broadly applicable to other settings beyond the single desk. If the reward function is indeed restricted to the specific desk environment, then scaling this approach to a wide-range of real-robot tasks would be very difficult.
>
> It is correct that the learned reward is trained on data from a single environment, and thus is likely restricted to that environment. Because it makes few assumptions about data sources, in principle LORL could train on large amounts of data from different environments, and as a result yield some environment generalization. However, there are not yet many manipulation datasets consisting of diverse behavior in many different environments. Moreover, even if the reward function did generalize well, generalization across environments is an open challenge in visuomotor control, making it difficult to evaluate such a reward function. As such, we leave this to future work.
>
> > What was the action-space used for data-collection and execution? SE-3? SE-2? Were there any hand-crafted heuristics for the exploration process? How long did the random-exploration data collection process take per task?
>
> The action space of both the simulated and real robot was 3DOF delta-end effector control. The data collection in simulation was using a uniform random policy (collection was fast since in simulation), and the data on the real robot was taken from the replay buffer of a previously trained agent (from an entirely different project). The reused real robot dataset from the other project corresponds to approximately 7 robot days total across all of the tasks.

---

> > ### Comment · Reviewer_aTgV · 2021-08-25
> > **Response to authors.**
> >
> > Thank you for the detailed response. I appreciate the clarifications and acknowledgments of the limitations.
> >
> > _while the paper used a random exploration policy to collect data in simulation, the real robot experiments used the replay buffer of a previously trained RL agent_
> > On one hand, this is great. Re-using datasets/trained-models is a common paradigm in vision and language, but not so much in robotics. So it's great to see that data from other frameworks like a pre-trained RL agent is re-used in this work.
> >
> > But on the other hand, it also makes the learning process very complicated. In order to reproduce the results of this work, an experimenter would need to reproduce several other works first. Without a huge collaborative effort, this would be difficult.
> >
> > Do the authors have any thoughts how to make this process more reproducible?

---

> > > ### Author Response · Authors · 2021-08-25
> > > **Re: Response to authors.**
> > >
> > > Thank you for the quick follow-up.
> > >
> > > We intend to release the full real robot dataset and corresponding crowdsourced language annotations. We have already begun efforts to prepare the dataset for release. We believe that releasing the annotated dataset will help the reproducibility of the real robot results to the extent possible. Of course, we agree that reproducing the real robot data collection process is difficult. In your analogy to vision and language, it is also difficult to reproduce the data collection process of, e.g., WikiText, if the creation of Wikipedia is considered part of that process. We believe that this difficulty is inevitable as datasets grow in size and variety, but also that this should not cause us to shy away from large-scale real-world validations of robotic learning methods.
> > >
> > > We also plan to release the simulated environments and code to enable reproducing and building on the algorithm in simulation.

---

### Official Review · Reviewer_k4Ao · 2021-07-23

**Originality:** Very Good
**Technical Quality:** Very Good
**Clarity Of Presentation:** Very Good
**Impact:** 4

**Recommendation:**

Strong Accept: I recommend accepting the paper and will argue for my recommendation even if other reviewers hold a different opinion.

**Summary:**

This work proposes a new approach to specify tasks using natural language. Motivated by the drawbacks of goal images, specifically over- and under-specification, the authors propose to instead represent reward using a language-conditioned classifier: does the transition between the initial and final states (images) satisfy the linguistic task description? In order to learn such a classifier, a dataset of state transitions and their corresponding natural language descriptions is required to ground the task description in the robot’s low-level observations. To scale this process, the authors use sub-optimal datasets that are later annotated with natural language. They show the proposed language-conditioned reward function can be combined with model-based RL in both real and simulated domains to successfully execute several tasks. Further, the authors analyze the performance of their method under different types of linguistic variations.


**Issues:**

The main issue with this paper is missing discussion about the limitations of the proposed reward specification method.

**Reviewer Expertise:**

Very good: Comprehensive knowledge of the area

**Strengths And Weaknesses:**

This paper addresses an important problem in robotics: that of task specification. Natural language is a good candidate as has been shown by previous works in the literature and the current paper. One strength of this paper is the insight to formulate the reward function as a classification problem. This type of weak supervision allows the authors to scale their data collection, while still being able to flexibly define a wide variety of tasks. Along with this proposal, the authors show the reward, when learned in this way, can be successfully optimized within a model-based RL framework. Another strength of this paper is the robot evaluations which show the feasibility of such an approach in the real world. In addition, this paper is well organized and the video was helpful as an overview of the approach. Along with these strengths, there are a few weaknesses that could be addressed to further improve the paper.

The proposed approach addresses some of the downsides of using goal images. While I agree this work can address the “task overspecification” problem, it is not immediately clear to me how the framework can be used to solve some non-goal reaching tasks (as alluded to in lines 37 or 171). The types of tasks that can be represented are limited by the reward being a function of the initial and final states. This seems to preclude some types of tasks such as those involving constraints on the trajectory (e.g., move a cup without spilling) or temporal (but not indefinite) tasks (e.g., turn the faucet three times). This paper would be strengthened by a discussion about the limitations of this type of task specification as well as clarification on how the model can be used to continuously rotate a faucet (line 171, is the planning approach repeatedly applied indefinitely to continue maximizing reward?).

In addition, I have included some minor comments and questions below regarding the clarity of the paper:
- Is there any intuition for why all the methods do worse on the “open door” task? (Figures 6 and 7)
- In the appendix, why are different visual planning models used for sim and real? This insight may be useful to those working on similar domains.
- It is mentioned that delta end-effector control is used, how far does each action move the gripper? How long is each trajectory typically?
- For the sim experiments, the data is collected using a random policy. How often does this policy lead to “interesting” data that corresponds to a task?


**Summary Of Recommendation:**

I recommend this paper be accepted to CoRL. It tackles the difficult problem of task specification and offers a useful perspective to the community. The results back up the claims of the paper and the methodology is well described.

---

> ### Author Response · Authors · 2021-08-24
> **Response to Reviewer k4Ao**
>
> Thank you for your positive feedback.  We respond to your main critiques individually below, and have made revisions to the paper (highlighted in *green*) based on your comments, which we believe has improved the paper. Please let us know if you have any remaining concerns or questions!
>
> > ...it is not immediately clear to me how the framework can be used to solve some non-goal reaching tasks (as alluded to in lines 37 or 171). The types of tasks that can be represented are limited by the reward being a function of the initial and final states. This seems to preclude some types of tasks such as those involving constraints on the trajectory (e.g., move a cup without spilling) or temporal (but not indefinite) tasks (e.g., turn the faucet three times)...clarification on how the model can be used to continuously rotate a faucet (line 171, is the planning approach repeatedly applied indefinitely to continue maximizing reward?).
>
> The reviewer is absolutely correct. We revised the paper to clarify that LORL can only capture tasks which are reflected in a change of state. Unlike single goal states, this can (a) capture tasks where many possible states (or changes in state) imply task completion, and can (b) be iteratively applied in a closed loop to perform task indefinitely without additional specification (e.g. the reward for “move right” is relative to the agent’s current position, so applying it iteratively will encourage the agent to continuously move right). LORL **cannot** be applied to tasks which are path dependent, e.g. “jump around in place”, and “move right slowly”. We expanded the discussion of limitations in the paper, including a discussion of using the full video clip in LORL as an extension in future work to handle such path dependent tasks.
>
> > Is there any intuition for why all the methods do worse on the “open door” task? (Figures 6 and 7)
>
> Opening the drawer is a more challenging task in terms of visuomotor control than the others, as it requires moving the arm precisely into the drawer handle/drawer cavity, then pulling open. When planning with an imperfect visual dynamics model, the robot may often narrowly miss the drawer handle and fail to open it.
>
> > In the appendix, why are different visual planning models used for sim and real? This insight may be useful to those working on similar domains.
>
> The real robot domain has more complex visuals and dynamics than the simulation environment, as well as having several cameras for full observability of the scene. As a result we use a more light-weight architecture in simulation, and a more expressive architecture on the real robot.
>
> > It is mentioned that delta end-effector control is used, how far does each action move the gripper? How long is each trajectory typically?
>
> On the real robot each action can move the gripper up to 7cm. The episode length on the real robot is 30 timesteps, and trajectories are planned 5 timesteps into the future. We have also added these details to the appendix.
>
> > For the sim experiments, the data is collected using a random policy. How often does this policy lead to “interesting” data that corresponds to a task?
>
> In the simulation data, the random exploration policy interacts with some object (generating an instruction that is not “do nothing”) in 40,108 out of 50,000 episodes. While this will vary depending on the environment and exact exploration policy, the key distinction of LORL is that it does not make any assumptions about the optimality of the actions taken in this offline data, allowing it to learn from data generated by a wide range of behavior policies, including
> 1. autonomous exploration data (e.g. random/scripted/intrinsically motivated)
> 2. replay buffers from previously trained RL agents
> 3. data collected by human experts (e.g. demonstrations/play data),
> 4. or even data which does not have action labels at all (for training the reward function).

---

> > ### Comment · Reviewer_k4Ao · 2021-08-30
> > **Response to authors**
> >
> > I would like to thank the authors for taking the time to respond to my concerns and provide clarifications. The added discussion about the scope of the learned reward function addresses my main concern.
> >
> > Regarding the clarification about the “open door” task, the ability to use other sources of data (not just random exploration) seems like a promising approach to enable both more complicated tasks and language. I encourage the authors to continue exploring the challenges associated with longer horizon tasks in their future work!

---

### Official Review · Reviewer_P4BX · 2021-08-02

**Originality:** Fair
**Technical Quality:** Fair
**Clarity Of Presentation:** Good
**Impact:** 3

**Recommendation:**

Weak Reject: I recommend rejecting the paper, but will not argue for my recommendation if the majority of other reviewers have a different opinion.

**Summary:**

The paper proposes an RL-based approach to language understanding that is able to map instructions to reward functions and the corresponding policy using only offline data. Given a set of potentially sub-optimal demonstrations in the form of state-action sequences, each paired with a crowd-sourced instruction, the proposed framework learns a binary reward as a classifier that is a function of initial and current states, as well as language. Using model-based RL, this reward is combined with the learned transition function to learn the corresponding policy in an offline fashion. The method is evaluated in simulation and shown to outperform recent RL-, behavioral cloning- and goal image-based language understanding baselines. The paper then presents results for a handful of tasks with a physical robot arm.

**Issues:**

* The paper incorrectly states that references [27] and [28] are limited to simulated robots
* The following work is relevant

Blukis, V., Terme, Y., Niklasson, E., Knepper, R. A., & Artzi, Y. (2019). Learning to map natural language instructions to physical quadcopter control using simulated flight. arXiv preprint arXiv:1910.09664


* Line 106: It is not clear from the text why learning from offline data enables the learning of language-conditioned policies on real robots, Can the authors elaborate?
* Is the space of language associated with each task assumed to be known?
* Lines 170--172: With regards to the faucet example: If the agent rotated the faucet by 10 deg and then moved it back and forth by 5 degrees, would it not look like it was rotated at each time step under this definiition, even though I wouldn't be rotating it.
* The positive selection strategy seems arbitrary. Why is this labeling scheme reasonable?
* The paper mentions that identifying negative examples is challenging, yet the proposed approach is quite simple.
* For the evaluations in Section 5.1, is there a difference between the instructions that were seen during training. The description of the zero-shot experiments in Section 5.2 suggests that the sentences were the same,
.

**Reviewer Expertise:**

Excellent: Expert knowledge on the topic of the paper

**Strengths And Weaknesses:**

STRENGTHS

+ The use of language is a compelling alternative to goal images in terms of specifying task goals. As the paper notes, there a number of limitations with requiring that goals be specified in terms of images, which are alleviated when using language as the goal specification.
+ The ability to learn language-guided policies entirely from offline demonstration has advantages over contemporary methods that learn reward functions from offline data, but then require on-robot learning.

WEAKNESSES

- Despite suggestions to the contrary, the approach seems limited to tasks that can be defined in terms of a single change in state vs. those that require reasoning over sequences of states (e.g., determining whether the agent has "turned the faucet to the left" requires reasoning over sequences of states and not a single state-pair).
- The paper should more clearly compare how this approach differs from related RL-based approaches to language understanding. While many of these do require that the resulting policies be learned online, they do not require access to state-action trajectories as is the case here, which can be expensive to collect as in the case requiring teleoperated demonstrations. Further, by learning policies in a model-free manner, they are not subject to the bounds on optimality posed by model-based RL.
- The paper should clarify which existing methods assume that the underlying demosntrations were provided by experts. This does not seem to be the case for symbol grounding-based approaches to language understanding, for which the grounding of the symbols to actions does not need to be optimal. Indeed the standard means by which these methods collect training data is to crowd-source annotations for existing demonstrations that need-not be optimal (e.g., see Tellex et al 2011).
- From a language understanding perspective, experimental evaluation is underwhelming. As noted below, the results seem more of an assessment of visuomotor control than of language understanding given (a) the simple nature of the instructions and (b) the limited evaluation of zero-shot performance.
- In its current form, the paper over-states the the method's utility on real robots. It is nice to see experiments beyond simulation, but the simplistic nature of the language suggests that the primary challenge here is on the visuomotor control rather than on the language understanding.



REFERENCES

[1] S. Tellex, T. Kollar, S. Dickerson, M. R. Walter, A. Banerjee, S. Teller, and N. Roy. Understanding natural language commands for robotic navigation and mobile manipulation. In Proceedings of the National Conference on Artificial Intelligence (AAAI), pages 1507–1514, San Francisco, CA, August 2011.


**Summary Of Recommendation:**

The idea of learning language-conditioned policies entirely from sub-optimal demonstrations is interesting. However, the instructions are much simpler than those considered by other language understanding methods and the generalizability is unclear. It is nice to see the method demonstrated on a physical robot, however it seems that this is more due to the visuomotor policy learning than it is language understanding.

UPDATE BASED ON AUTHOR RESPONSE

I thank the authors for their detailed response, which resolved some of the questions and concerns that I initially raised in my review.

I like the overall motivation, but the significance of the contributions aren't clear. This is not the first paper to use a classifier to define a reward function (see below). I agree with the authors and other reviewers on the importance using language as a means of specifying abstract goals (with much of the work in language understanding focused on language that provides a finer grained specification of goals). However, as the authors note, the language here is rather simple. It seems that the primary contribution is the ability to learn visuomotor policies offline with language as an alternative images for specifying goals (i.e., language is sort of a second-class citizen). That's not to say that the contributions aren't sufficient, but that the paper should make them clear.


[2] Konidaris et al., Robot Learning from Demonstration by Constructing Skill Trees, IJRR 2012

[3] Kulkarni et al., Hierarchical Deep Reinforcement Learning: Ingegrating Temporal Abstaction and Intrinsic Motivation, NeurIPS 2016

---

> ### Author Response · Authors · 2021-08-24
> **Response to Reviewer P4BX (1/2)**
>
> Thank you for your valuable feedback. We respond to your main critiques individually below, and have made revisions to the paper (highlighted in *green*) based on your comments, which we believe has improved the paper. Please let us know if you have any remaining concerns or questions!
>
> > Despite suggestions to the contrary, the approach seems limited to tasks that can be defined in terms of a single change in state vs. those that require reasoning over sequences of states (e.g., determining whether the agent has "turned the faucet to the left" requires reasoning over sequences of states and not a single state-pair).
>
> We revised the paper to clarify that LORL can only capture tasks which are reflected in a change of state. Unlike single goal states, this can (a) capture tasks where many possible states (or changes in state) imply task completion, and can (b) be iteratively applied in a closed loop to perform task indefinitely without additional specification (e.g. the reward for “move right” is relative to the agent’s current position, so applying it iteratively will encourage the agent to continuously move right). LORL **cannot** be applied to tasks which are path dependent, e.g. “jump around in place”, and “move right slowly”. We expanded the discussion of limitations in the paper, including a discussion of using the full video clip in LORL as an extension in future work to handle such path dependent tasks.
>
> > The paper should more clearly compare how this approach differs from related RL-based approaches to language understanding. While many of these do require that the resulting policies be learned online, they do not require access to state-action trajectories as is the case here, which can be expensive to collect as in the case requiring teleoperated demonstrations.
>
> We have added further discussion of these prior works to the related work section of the revision. One key difference is that for vision-based robot manipulation on a real robot, the samples needed to learn from language via online RL can be prohibitively time consuming, which motivates our fully offline approach. Moreover, we clarify that this paper does *not* use expert demonstration data, but instead leverages suboptimal, autonomously-collected offline datasets, and can learn the reward component even from data without action labels. This enables the method to learn from more scalable data sources compared to prior works.
>
> > The paper should clarify which existing methods assume that the underlying demosntrations were provided by experts. This does not seem to be the case for symbol grounding-based approaches to language understanding, for which the grounding of the symbols to actions does not need to be optimal. Indeed the standard means by which these methods collect training data is to crowd-source annotations for existing demonstrations that need-not be optimal (e.g., see Tellex et al 2011).
>
> Thank you for pointing this out - we have corrected the related work section to clearly describe which methods use expert provided demonstrations. Namely the symbol grounding approaches (like Tellex et al), do not depend on demonstrations, but rather directly ground language to symbolic states/actions that are executed in the environment.
>
> > From a language understanding perspective, experimental evaluation is underwhelming. As noted below, the results seem more of an assessment of visuomotor control than of language understanding given (a) the simple nature of the instructions and (b) the limited evaluation of zero-shot performance. In its current form, the paper over-states the the method's utility on real robots. It is nice to see experiments beyond simulation, but the simplistic nature of the language suggests that the primary challenge here is on the visuomotor control rather than on the language understanding.
>
> You are correct that the focus of this work is not on complex language understanding, but rather on more effective task specification for visuomotor robotic manipulation. We agree that progress must be made on both more complex grounded language understanding and visuomotor control policies before robots will be useful in the real world. We have toned down some of the claims in the revision, specifically in Section 5.3 and Section 6, regarding the methods overall utility.
>
> > The paper incorrectly states that references [27] and [28] are limited to simulated robots. The following work is relevant - Blukis, V., Terme, Y., Niklasson, E., Knepper, R. A., & Artzi, Y. (2019). Learning to map natural language instructions to physical quadcopter control using simulated flight. arXiv preprint arXiv:1910.09664
>
> Thank you for catching that! We have corrected the references and cited the above work in the revision.

---

> > ### Author Response · Authors · 2021-08-24
> > **Response to Reviewer P4BX (2/2)**
> >
> > > Line 106: It is not clear from the text why learning from offline data enables the learning of language-conditioned policies on real robots, Can the authors elaborate?
> >
> > Online deep reinforcement learning on a physical robot can be time consuming, costly, and unsafe. Hence the ability to learn from previously-collected, offline data is particularly valuable for real robots, as it reduces the need for online interaction. For example, the research and experiments in this paper did not involve any real robot interaction for learning, as the models were trained by reusing an offline dataset collected in a previous paper. Only the evaluations required roll-outs on the physical robot.
> >
> > > Lines 170--172: With regards to the faucet example: If the agent rotated the faucet by 10 deg and then moved it back and forth by 5 degrees, would it not look like it was rotated at each time step under this definiition, even though I wouldn't be rotating it.
> >
> > We agree that LORL cannot handle path-dependent tasks, and have clarified the paper accordingly.
> >
> > > The positive selection strategy seems arbitrary. Why is this labeling scheme reasonable?
> >
> > The positive selection (using an episode’s initial/final states and corresponding annotation as positive) is based on the assumption that when a human annotator described a video with a particular instruction, they are doing so correctly. The noisy positives strategy is to create more positive examples, and is based on the (usually true) assumption that the labeled annotation for a video is also valid in many subsequences of the original video. The hyperparameter alpha is used to tune how many subsequences to consider valid for the given annotation. We have also since run an ablation on the real robot and found removing noisy positives on the robot reduces performance (average success rate 66% -> 32%), suggesting that with limited data, generating additional positives with the noisy labeling is important.
> >
> > > For the evaluations in Section 5.1, is there a difference between the instructions that were seen during training. The description of the zero-shot experiments in Section 5.2 suggests that the sentences were the same,
> >
> > Correct, the instructions in Section 5.1 are seen instructions in the training data, while in Section 5.2 the instructions are unseen. We have clarified this in Section 5.1.

---

### Meta-Review · Area_Chair_6JQN · 2021-08-13

**Recommendation:** Accept (Poster)
**Confidence:** 4

**Metareview:**

All reviewers agree that the presented approach is interesting and
that natural language descriptions pose compelling alternatives to goal images wrt task goal specification.

The paper would benefit from a discussion regarding limitations and scalability (especially wrt to real world applications).
Furthermore some claims and statements, as pointed out by the reviewers, should be clarified and referenced (where appropriate).

Most of the criticism can be addressed trough clarifications.

## Post-Rebuttal Update:
The authors successfully addressed and clarified the reviewers concerns and incorporated their feedback into an improved version of the manuscript.

---

### Decision · Program_Chairs · 2021-09-13

**Decision:**

Accept (Poster)

**Comment:**

All reviewers agree that the presented approach is interesting and
that natural language descriptions pose compelling alternatives to goal images wrt task goal specification.

The paper would benefit from a discussion regarding limitations and scalability (especially wrt to real world applications).
Furthermore some claims and statements, as pointed out by the reviewers, should be clarified and referenced (where appropriate).

Most of the criticism can be addressed trough clarifications.

## Post-Rebuttal Update:
The authors successfully addressed and clarified the reviewers concerns and incorporated their feedback into an improved version of the manuscript.